# The Financial/Accounting Impact of FFP on Participating in European Competitions: An Analysis of the Spanish League

Alberto Calahorro-López [1] , Melinda Ratkai [2] and Julio Vena-Oya [3,*]

1   Department of Financial Economics and Accounting, Faculty of Social and Legal Sciences, University of Jaén, 23071 Jaén, Spain
2   Department of Finance and Management, Les Roches Education, 29602 Marbella, Spain
3   Department of Business Organisation, Marketing and Sociology, Faculty of Social and Legal Sciences, University of Jaén, 23071 Jaén, Spain
*   Correspondence: jvena@ujaen.es

**Abstract:** This paper analyses the impact of Financial Fair Play (FFP) on clubs' finances and on the relationship between them and clubs' sporting outcomes in the Spanish league. To this end, financial ratios and accounting variables obtained from the clubs' own annual accounts, published from 2004 to 2019, are analysed, and the Mann–Whitney test is used to describe which differences are significant. The objective is threefold: firstly, we describe the financial/accounting structures of Spanish league football clubs, showing how both their financial statements and ratios have evolved after the application of this law, providing evidence of whether FFP is an adequate tool to guarantee the long-term viability and sustainability of football clubs, as intended by the UEFA. Secondly, we show the relationship between financial/accounting performance and sporting results. Thirdly, the paper looks at whether FFP has impacted the gap between the top clubs and the rest. The results show that, after the implementation of FFP, clubs' financial/accounting health has improved, and a change in their efficiency in this regard can be observed. Although FFP is a tool that achieves the objective for which it was created, the possibility that the gap between the elite clubs and the rest may be increasing should not be ignored.

**Keywords:** finance; economic impact; accounting; Financial Fair Play; Spanish league; UEFA; sports profitability

## 1. Introduction

Since the 1990s, the budgets of professional clubs in European football have been rising, largely due to surging broadcasting and merchandising revenues, foreign investors, fan growth, and the expansion of the European football community (Birkhäuser et al. 2019). Currently, it is the "king of sports", and the organization of its competitions has become a major industry, generating billions of dollars for club owners (Kalashyan 2021). However, the football industry has, surprisingly, posted persistent losses at Europe's elite clubs in France, Spain, Italy, England and Germany (Andreff 2007; Ascari and Gagnepain 2007; Dietl and Franck 2007; Lago et al. 2004). Although their revenue has increased substantially since the 1990s, their expenses (mainly transfer payments and salaries) have outpaced these increases (Ascari and Gagnepain 2007; Barajas and Rodríguez 2014; Franck 2014), with the net losses of the 734 European member clubs rising 760% in the five-year period from 2006 to 2011 (Franck and Lang 2014; Francois et al. 2021).

Despite the fact that the first legislation began to try to remedy the financial instability of clubs in Europe as early as the 1980s (Italy in 1981 and Spain, with Law 10/1990 of 15 October on Sport), the literature is consistent in that most of the clubs in the "big five" leagues have had financial problems (Ascari and Gagnepain 2007; Barajas and Rodríguez 2010, 2014; Dimitropoulos and Koumanakos 2015; Solberg and Haugen 2010; Szymanski and Smith 1997; Vrooman 2007), especially after the Bosman ruling (Ascari and Gagnepain

2007; Solberg and Haugen 2010; Vrooman 2007). In December 1995, the European Court of Justice (ECJ) ruled that the restriction on the transfer of out-of-contract players is contrary to the provisions of the European Economic Community. Since this decision, out-of-contract players can move freely from one team to another as long as both teams are established in the EEC.

After the Bosman Act, clubs engaged in a race to acquire talent, with this struggle being the main cause of financial instability (Ascari and Gagnepain 2007; Barajas and Rodríguez 2010, 2014; Dimitropoulos and Koumanakos 2015; Solberg and Haugen 2010; Szymanski and Smith 1997; Vrooman 2007). As a result, it can be observed that the most common form of financing for football clubs was through debt (Barajas and Rodríguez 2010, 2014; Dimitropoulos and Koumanakos 2015; Drut and Raballand 2012; Prigge and Tegtmeier 2019) with high leverage, a weak cash generation capacity, and low levels of ROA and ROE (Dimitropoulos and Koumanakos 2015). In a study of the Italian league, it was found that wage expenses amounted to 53% of total revenues, on average (Dimitropoulos and Scafarto 2021), ratifying previous findings that players' salaries are the most significant expenses of Italian football clubs (Nicoliello and Zampatti 2016; Regoliosi 2016). This financial instability coincides with increasing revenues from TV rights and other sources, leading to an inefficient relationship between revenues and expenses (Ascari and Gagnepain 2007; Solberg and Haugen 2010). That is, European clubs tend to overinvest in players and have financial problems, despite generating high revenues, because they prioritise sporting objectives over organizational or financial ones (Solberg and Haugen 2010).

In 2010, the UEFA announced a set of rules, dubbed "Financial Fair Play (FFP) Regulations", aiming to improve the discipline and soundness of the management of European football clubs, decrease the pressure on salaries and player acquisition (transfer fees), and protect the long-term viability of European club football (Birkhäuser et al. 2019). UEFA has targeted improved financial positions and performance by balancing revenues and expenses to reach, at least, a break-even point (Morrow 2013, 2014). Failure to comply with the financial requirements of this new law automatically leads to sanctions and a loss of revenue, placing the financial viability of most clubs at risk (Dimitropoulos 2016).

This law is expected to serve as a mechanism to monitor the expenses of players and thus compensate for clubs' weak governance structures. FFP was also intended to restore incentives for efficient management (Franck 2014; Sass 2016) by limiting the possibility of weakly managed clubs to be rescued by their owners (Dimitropoulos and Scafarto 2021). Nonetheless, the scientific evidence as to the effects of FFP and whether it is driving a shift towards more sustainable business models continues to be an empirical question (Rohde and Breuer 2018). There is consensus that FFP has succeeded in improving the financial health of European clubs (Ghio et al. 2019), but unintended effects have also arisen, such as an increase in imbalances (from a sports achievement perspective) between national and international leagues (Gerhards and Mutz 2017; Ramchandani et al. 2018).

The question arises: what can clubs do to achieve their sporting objectives (that is, to compete at the European level)? They can, firstly, increase the talent on their football teams (Rohde and Breuer 2016).

According to Maqueira et al. (2019), there are two ways for football clubs to achieve this: internal talent development or external talent attraction. The development of internal talent has been used, historically, by F.C. Barcelona ever since the arrival of Johan Cruyff in the late 1980s. However, at the end of the last century, F.C. Barcelona endured one of the worst periods in its history, suffering one of its most acute economic and sporting crises. From the 1998–1999 season until 2004–2005 (five seasons), it won no major titles, and it was not until the acquisition of Ronaldinho Gaúcho from Paris Saint Germain, in the 2003–2004 season, that the economic and sporting foundations of what was to prove the most successful period in the club's history were cemented. This, together with the internal blossoming of young players such as Carles Puyol, Andrés Iniesta, Xavi Hernández, Víctor Valdés, Pedro, and Sergio Busquets, among others, and the arrival of Lionel Messi, allowed Barcelona to surpass Real Madrid and Manchester United in revenue for the first

time. Figure 1 shows how FC Barcelona, with this hybrid model based on the internal development of players and the acquisition of external talent (mainly with the arrival of Ronaldinho Gaúcho), managed to rise from seventh to first place in revenue (Deloitte 2005, 2020). In addition, it achieved unprecedented dominance in the club's history in both Europe and the Spanish league, winning 10 of 16 Spanish championships, 4 Champions Leagues, and 6 King's Cups, among others.

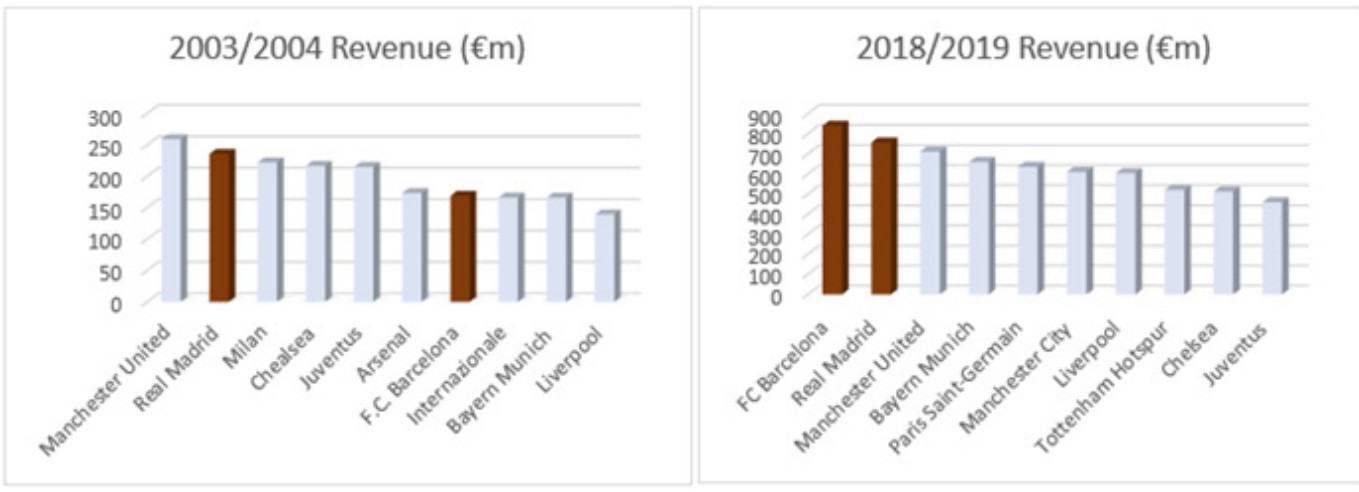

**Figure 1.** Evolution of the top 10 clubs in terms of revenue.

This study aims to empirically explore the above arguments in the context of the Spanish Premier League. With the objective of examining the impact of FFP on the Spanish professional football league at both the sporting and financial accounting levels, this paper poses a series of research questions that are set forth in the following section. The paper is relevant because it adds to the literature on the impact of FFP on the financial/accounting structure of Spanish league football clubs, showing how both the accounting statements and the financial ratios published by the clubs themselves have evolved following the implementation of this law. Secondly, it provides evidence on whether FFP is an adequate tool to guarantee the long-term viability and sustainability of football clubs. Thirdly, we will show the relationship between financial/accounting performance and sporting results. In addition, the paper shows whether FFP has influenced the gap between top clubs and the rest. Furthermore, the paper addresses whether FFP has influenced the competitive balance of the Spanish league by examining the difference between two groups of clubs: those that participate in European competition and those that do not.

The rest of the paper is structured as follows. The Section 2 presents a comprehensive discussion of the previous literature. The Section 3 describes the data selection procedure and the study's design. In Section 4 the main empirical results are discussed, while the Section 5 concludes the paper and cites its different implications.

## 2. Literature Background

The importance of efficient management and the search for competitive balance in sporting competitions has been the subject of studies since the 1950s and is beneficial both for the clubs and competition itself (Atkinson et al. 1988; El-Hodiri and Quirk 1971; Neale 1964; Rottenberg 1956). Preventing talent from moving to a largely impenetrable set of elite clubs, particularly in the football industry, has been a major issue for decades. This balance could be achieved by regulating a transfer system and the labour market for players, thereby preventing talent from moving to a reduced group of clubs (Sloane 1969). Other publications conclude that the redistribution of revenues among competing clubs is the solution (Dabscheck 1975; Davenport 1969), an argument that was disputed previously by Rottenberg (1956), who suggested that, if finances were distributed equally, no club would have an incentive to win.

The finances of the football industry, specifically the accumulation of debt by European clubs, is an increasing cause for concern among football authorities (Drut and Raballand 2012), as it could affect the competitive balance of competitions and the viability of the industry. This competitive balance distinguishes the North American model, which prioritises financial objectives, from the European model, which gives precedence to sporting objectives (Dimitropoulos and Koumanakos 2015; El-Hodiri and Quirk 1971; Solberg and Haugen 2010), which could explain the unstable situation of European clubs.

As the literature is not unanimous on the relationship between sporting performance and financial management, this paper presents a comprehensive review of the subject. Elite clubs, or clubs that have achieved better sporting results, have been less financially efficient (Barros and Leach 2007; Guzmán and Morrow 2007), as there seems to be a direct link between investment in players and success on the pitch (Rohde and Breuer 2016; Szymanski 2017). The relationship between player salaries and sporting performance is extremely robust and has been extensively documented (Drut 2011; Pedace 2008; Szymanski and Kuypers 1999), as well as the fact there exists a relatively stable and predictable relationship between team performance and revenues earned (Szymanski 2017). This correlation between talent and sporting success could explain the aforementioned "race for talent" in the quest to meet sporting objectives by increasing spending and debt, thereby aggravating financial problems (Barros and Leach 2006; Dimitropoulos and Koumanakos 2015). In this sense, those clubs with continuous deficits due to overspending on their squads "infect" others who, trying to keep up with the competition, increase their spending on talent and their exposure to risk to unsustainable extents (Müller et al. 2012).

These continuing deficits have led to insolvency problems in the football industry (Szymanski and Weimar 2019) and a situation of financial instability such that if the Spanish, English and Italian leagues were normal companies, they would be threatened with financial bankruptcy (A.T. Kearney 2010; Schubert 2013). Nevertheless, European football has a high "survival rate", and clubs rarely disappear from the market, despite high levels of debt (Kuper and Szymanski 2009; Schubert 2013). In the Spanish case, the situation was critical, with a serious structural deficiency and a worrying debt ratio, indicating that most clubs were, technically, insolvent (Barajas and Rodríguez 2010). More than half of the First and Second Division clubs were in bankruptcy proceedings, or on the verge of bankruptcy, with a severe lack of credibility in the market (Gómez 2017). It should be noted that Spanish football clubs do not benefit from the full access to finance used by multinationals and medium-sized firms, which could be due to the fact that a cultural tendency towards uncertainty avoidance proves to be a factor hampering firms' access to all types of finance (Aggarwal and Goodell 2014). Current stock market returns are weaker for countries with higher levels of uncertainty avoidance, while in more open markets, they are associated with relatively high levels of foreign participation in the stock market (Tsalavoutas and Tsoligkas 2021). This fact might explain why Spanish league clubs have not yet tapped the stock markets. Therefore, our analysis is based on the Spanish football league. Along with this financial deficiency, investors started to appear and external resources increased, causing lopsided competitive levels favouring just a few clubs, such that cases of "financial doping" could be occurring (Olsson 2011).

This regular practice of financing clubs in order to maximize sporting objectives began to worry the UEFA, which considered this financial "short-termism" a threat to the stability of European football (Kalashyan 2021). Reacting to this, in 2010, UEFA introduced "Financial Fair Play Regulations", going into effect in the 2011–2012 season. This is a set of rules limiting each club's spending according to the financial resources they are able to generate. The main objective is to reduce the growing indebtedness of European clubs and their increasing dependence on investors. After FFP, external funding and "equity investment from rich benefactors cannot be counted as part of the club's income" (Schubert 2013; Serby 2016). The aim was for financial efficiency to become a decisive factor for organizational success (Barros et al. 2014). Clubs that do not meet certain financial indicators are obliged to draw up budgets detailing their future strategic plans. Under the

regulations, the UEFA also supervises and enforces financial control and has the authority to impose sanctions, which can even involve disallowing football clubs from participating in European competitions and buying or selling players for a certain period of time, or the application of financial penalties (Morrow 2014). Due to the fact that UEFA is the organizing entity for European competitions, clubs naturally strive to comply with its rules.

There were several reactions to this law, with major European football stakeholders, such as the European Club Association, the Association of European Football Leagues, the players' union, and the European Commission declaring their support for UEFA's proposals (Schubert 2014). There was also a change in the landscape of the European football industry, as well as exponential growth in terms of academic work studying this law in one way or another (Calahorro-López and Ratkai 2022).

*The Impact of FFP*

A number of studies from a financial/accounting point of view have arisen as a result of this law. One of them observes that FFP protects systemic financial stability and seeks to ensure the survival of an open league system (Drut and Raballand 2012; Franck 2014), with these scholars focusing on the five biggest leagues (the Spanish, German, English, French, and Italian leagues) and highlighting the importance of regulation, since those clubs that are allowed to incur larger deficits, mainly in order to boost the quality of their squads (both in intangible assets and players' salaries), obtain better sporting results. Leagues with more regulation, such as the German and French leagues, have had worse results in European competitions. The paper concludes that FFP would, theoretically, help to equalize the competitive balance, while also warning that it could widen the gap between a small group of big clubs and the rest.

There is a certain consensus that the application of FFP improves the financial situations of clubs (Bernile and Lyandres 2011; Terrien et al. 2017) with better ROAs (Dimitropoulos 2016) and cash flows (Dimitropoulos and Koronios 2018). FFP has, for example, had a positive impact on the profitability of Italian clubs (Ghio et al. 2019). Dimitropoulos and Scafarto (2021) show that spending on salaries is 53% of total revenues, and that after FFP, the league remains highly leveraged (0.88) on average. Despite these data, it can be concluded that FFP dissuades clubs from joining the race to invest in talent and incentivizes greater financial efficiency through, among other things, net profits on the sale of transfer players. Post FFP, a relationship is observed between financial performance and sporting results and rankings (Galariotis et al. 2018), along with one between the market values of teams' players and their competitiveness (Birkhäuser et al. 2019).

FFP does not only have positive effects; Franck (2014) argues that the effects of FFP could actually undermine its stated objectives. Some authors argue that after the FFP's entry into force, the imbalance between the major European leagues has only increased (Birkhäuser et al. 2019; Terrien et al. 2017). Limiting the injection of financial funds from majority owners could mean that talent previously spread out (Madden 2015) could be favouring clubs that benefited from the previous lack of regulation (Freestone and Manoli 2017). In addition, these liquidity constraints conflict with France's DNCG laws, which were created to save French clubs from insolvency (Dermit-Richard et al. 2019).

From a financial point of view, more research is needed into the debate on the effects of FFP on teams' financial health (Franck 2018; Francois et al. 2021) and its impact on a number of financial items, such as cash flows and earnings (Dimitropoulos and Koronios 2018; Mareque et al. 2018).

Motivated by the above, the paper raises and answers the following research questions: (1) whether FFP has improved the financial health of Spanish league football clubs, (2) whether FFP is changing the business model of Spanish clubs with the aim of making them more financially efficient, and (3) whether there are differences before and after the implementation of FFP in the finances of clubs that participate in European competition and those that do not, which, according to the literature, could be affecting the competitive balance between them.

## 3. Data and Methodology

### 3.1. The Dataset: The Spanish League

Section 1 touches on why the Spanish league was selected, but in order to expand on this, Figure 1 was prepared, which shows the evolution of the top 10 clubs in terms of revenue. Here, it can be seen that the main Spanish clubs' positions have changed drastically over the period of time in question. We chose the Spanish league as the focus of our research in order to better understand the underlying reasons for this dramatic change.

Another aspect that is striking about Spanish soccer compared to other leagues (i.e., the English Premier League) is that it does not use the stock market as a source of financing. In addition, despite not being present on the stock market, FC Barcelona and Real Madrid have increased their revenues drastically and become the most successful clubs in Europe from a financial point of view. In Spain, only one club, Intercity (León 2021), has a stock market presence.

### 3.2. Data Selection Procedure

This study uses a set of financial and sporting data obtained from the SABI database and inspired by the methodology of Dimitropoulos and Scafarto (2021). Hence, the search was restricted to the following:

1. Clubs that participated in the First Division of the Spanish league during the period from 2004 to 2019 (clubs competing in the top division are more likely to participate in European competitions and are those most affected by the FFP regulation).
2. Clubs that have participated in at least 13 seasons in the First Division were selected.

As a result, a total of 13 clubs were selected, with the particularity that all of them have participated at least once in European competition.[1] These clubs are Athletic Club de Bilbao, Club Atlético de Madrid, F.C. Barcelona, Real Betis Balompié, Real Club Deportivo de la Coruña, R.C.D. Español, Getafe C.F., Málaga C.F., Real Madrid C.F., Real Sociedad de Fútbol, Sevilla F.C., Valencia C.F., and Villareal C.F.

### 3.3. Research Design

This paper analyses the impact of FFP regulation by proposing research questions on the financial health indicators most commonly used by the scientific community to measure the effects of FFP in the field of finance and accounting. To this end, two steps were taken. First, the differences between the means of the financial and accounting criteria were tested in order to analyse possible changes in the financial health of Spanish football before FFP (BFFP) and post FFP (PFFP).

With the aim of verifying whether there is a relationship between sporting and financial performance, the next step was to determine whether there are differences between the financial and accounting items of clubs that participated in European competition in the given year (EC) and those that did not (NEC). The differences were analysed in terms of BFFP and PFFP. Subsequently, we analysed whether the differences between groups of clubs increased or decreased after the introduction of FFP.

The data organized in the abovementioned way did not follow a normal distribution. So, in order to study the possible differences, a non-parametric test, the Mann–Whitney U (Mann and Whitney 1947), was used. This non-parametric test seeks to determine whether there are significant differences between two subsamples of a population. This population should follow a normal distribution, but the test is quite robust when the sample is larger than 30 individuals (Dowdy et al. 2011). In our case, the population is divided into two sub-samples, before and after FFP, to determine whether there were differences in clubs' finances in the wake of the introduction of this new regulation. Other studies on finance have used this technique to test for differences between two sub-samples (e.g.,: Olivier et al. 2018; Yang et al. 2019; Kusano 2020). Specifically, the Mann–Whitney test performs the following hypothesis test:

$$H_0 : \mu_i = \mu_j$$

$$H_1 : \mu_i \neq \mu_j$$

Adapting the Mann–Whitney test to our case, the hypotheses would be formulated as follows:

**H0:** *The samples come from the same distribution. (a.k.a. There is no evidence that FFP was related to the changes. In other words, the clubs before and after 2012 are not significantly different, either in terms of their finances or sporting achievements).*

**H1:** *The samples do not come from the same distribution. (a.k.a. There is statistical evidence that FFP was related to the changes. In other words, the clubs before and after 2012 were significantly different, statistically, in terms of their finances and/or sporting achievements).*

*3.4. The Test Variable and Grouping Criteria*

The test variables, as described in the previous section, were financial ratios and accounting items, such as Earnings Before Interest and Taxes (EBIT); Earnings Before Interests, Taxes, Depreciations and Amortizations (EBITDA); Earnings Before Taxes (EBT); Equity; Number of Employees (Employees); Indebtedness (Indeb); Intangible Assets (IA): Leverage (Lev); Net Turnover (Net tur); Operating Income (Oper. Income); Personnel Expenses (P.Expe); Personnel Expenses/Operating Income (PE/OI); Average Personnel Expenses (PEA); Net Profit/(Loss) for the period (Profit); Profit Margin (Prof. Marg.); Registered Capital (R. Capital); Return on Capital Employed (RCE); Return on Assets (ROA); Return on Equity (ROE); Return on Sales (ROS); Solvency Ratio (Solvency); Ratio of Financial Autonomy (SRFA); Total Assets (T.Assets). Further information about the calculations can be obtained from Appendix A.

There are two grouping criteria. The first grouping criteria is the FFP (financial aspect), which creates two groups: BFFP = 0, 2004–2011, PFFP = 1, 2012–2019). The second grouping criteria compares the differences between groups of clubs: those that participated in European competition (EC) = 1 and those that did not (NEC) = 0 (sporting objective aspect).

**4. Results and Discussion**

*4.1. Analysis of the Spanish League*

Table 1 shows the mean before FFP (BFFP) and post FFP (PFFP), differences between them, and the Mann–Whitney test, indicating whether the null hypothesis is rejected. Column 1 shows the items, financial ratios, and accounting items most commonly used in similar studies. As can be seen in Column 2 BFFP, which shows the averages from the 2004 until the 2011 season, the following table shows that the average values (EBIT, EBT, Profit, ROA, ROE, ROS, SRFA) for the relevant seasons were negative. Both indebtedness and leverage were very high, indicating that the most common form of financing for Spanish league clubs was through debt, in line with the studies by Barajas and Rodríguez (2010), Barajas and Rodríguez (2014), Dimitropoulos and Koumanakos (2015), Drut and Raballand (2012), and Prigge and Tegtmeier (2019). Low levels of ROA and ROE were also observed, coinciding with the research results of Dimitropoulos and Koumanakos (2015), El-Hodiri and Quirk (1971), and Solberg and Haugen (2010). A low solvency ratio indicates a low debt repayment capacity. The personnel expenses over operating income (PE/OI) reach 76.20%, which indicates that it has the greatest impact on income, in line with the pattern in the Italian league (Dimitropoulos and Scafarto 2021). Taking this fact together with the high levels of intangible assets, it could be stated that the cause of the poor financial health of the Spanish league's clubs is mainly the contracting of high-level players (Barajas and Rodríguez 2010).

**Table 1.** Comparing variables before and post FFP.

| ITEM | MEAN | | | Mann–Whitney | |
|---|---|---|---|---|---|
| | BFFP | PFFP | Absolute Dif. | Sig | Null |
| CashFlow | 16,168,752.02 | 36,683,436.00 | 20,514,683.99 | 0.000 | REJECT |
| EBIT | −8,018,639.11 | 13,301,529.27 | 21,320,168.37 | 0.000 | REJECT |
| EBITDA | 9,036,426.06 | 42,690,271.26 | 33,653,845.20 | 0.000 | REJECT |
| EBT | −10,587,822.65 | 9,694,114.91 | 20,281,937.56 | 0.000 | REJECT |
| Equity | 15,738,755.02 | 57,416,971.38 | 41,678,216.36 | 0.001 | REJECT |
| Employees | 163.31 | 305.46 | 142.15 | 0.000 | REJECT |
| INDEB | 111.40 | 104.04 | 7.36 | 0.007 | REJECT |
| Int Assets | 47,376,647.70 | 85,007,951.67 | 37,631,303.98 | 0.073 | |
| LEV | 1,348.79 | 224.43 | 1,124.37 | 0.100 | |
| Net tur. | 61,473,655.63 | 148,054,965.27 | 86,581,309.64 | 0.000 | REJECT |
| Oper. Income | 65,985,443.72 | 155,832,444.57 | 89,847,000.85 | 0.000 | REJECT |
| P. Expe. | 43,192,013.27 | 94,412,655.82 | 51,220,642.55 | 0.000 | REJECT |
| PE/OI | 76.20 | 68.52 | 7.68 | 0.014 | REJECT |
| PEA | 279.71 | 263.82 | 15.90 | 0.645 | |
| Profit | −886,313.15 | 7,370,680.40 | 8,256,993.5 | 0.000 | REJECT |
| Prof. Marg. | −34.23 | 4.09 | 38.32 | 0.000 | REJECT |
| R. capital | 17,986,560.45 | 56,491,938.32 | 38,505,377.86 | 0.000 | REJECT |
| RCE | 36.43 | 62.27 | 25.84 | 0.000 | REJECT |
| ROA | −13.06 | 6.17 | 19.22 | 0.000 | REJECT |
| ROE | −133.54 | 18.25 | 151.79 | 0.001 | REJECT |
| ROS | −88.48 | 4.42 | 84.06 | 0.000 | REJECT |
| Solvency | 0.67 | 0.65 | 0.02 | 0.609 | |
| SRFA | −11.41 | −4.05 | 7.36 | 0.007 | REJECT |
| T Assets | 200,664,540.11 | 291,107,116.59 | 90,442,576.48 | 0.124 | |

The Spanish clubs' poor financial results could also be explained by two facts: firstly, the lack of existing BFFP regulations (Ascari and Gagnepain 2007) and, secondly, the European focus on prioritising sporting objectives over and above organizational or financial ones (Dimitropoulos and Koumanakos 2015; Solberg and Haugen 2010; Szymanski and Smith 1997; Vrooman 2007). Therefore, despite the fact that the Spanish league is one of the biggest revenue earners, it has not been able to translate the revenue growth experienced since the 1990s into profits (see Barajas and Rodríguez 2014; Birkhäuser et al. 2019). Therefore, before the introduction of FFP, we can affirm that the Spanish football industry was in a situation of financial instability, corroborating the assertion in the report by A.T. Kearney (2010) indicating that if the Spanish, English, and Italian leagues were normal companies, they would be threatened with financial bankruptcy.

As can be seen in Table 1, Column 3, PFFP, there was a radical change in the financial-accounting data of the clubs following the implantation of FFP. Absolutely all the items and ratios improved. In addition, through the Mann–Whitney U test, we can identify which of these improvements are significant: cash flow, EBIT, EBITDA, EBT, equity, number of employees, indebtedness, net turnover, operating income, personnel expenses, PE/OI, profit, profit margin, registered capital, RCE, ROA, ROE, ROS, and SRFA. The results confirm similar research on major European leagues; it can be observed that after the implementation of FFP, the financial health of the Spanish league has significantly improved, as is the case in the rest of the major European leagues (Bernile and Lyandres 2011; Dimitropoulos 2016; Dimitropoulos and Koronios 2018; Dimitropoulos and Scafarto 2021; Terrien et al. 2017). It should be noted that, despite the evident improvement, there is still high leverage and indebtedness, as in the Italian league. After the implementation of FFP, we can state that FFP positively influenced the profitability of the clubs, in accordance with Ghio et al. (2019) and Gómez (2017).

In accordance with the results, we can answer Research Question 1: has FFP influenced the financial health of football clubs in the Spanish league? After observing an improvement

in all the financial measurements considered, there are indications that FFP has indeed positively impacted the financial health of Spanish league football clubs.

Regarding Research Question 2, as to whether FFP is rendering the business model of Spanish clubs more financially efficient, after FFP clubs increased their investment in talent, as evidenced by an increase in both intangible assets from 47,376,647.70 EUR to 85,007,951.67 EUR (although the difference is not significant) and personnel expenses from 43,192,013 EUR to 94,412,655 EUR (which is significant), though the PE/OI ratio decreased from 76.20% to 68.52% (which is also significant). Taking all this together with the improvement in the remaining criteria, it can be stated that a change in the financial management of these soccer clubs is evident. That is, investment in talent is greater, but financing through PFFP debt is no longer possible due to the limitations of the law itself, so there is a better correlation between income and expenses, and new sources of financing and financing based on increases in PFFP debt have disappeared. Dimitropoulos and Scafarto (2021) explain this by the fact that FFP pushes clubs from a management model based on "the race" to invest in talent by incurring debt towards one of greater financial efficiency, owing, among other things, to the net profits from the sale of transferred players in the Italian league. There is, therefore, evidence to be able to answer Research Question 2.

In view of the results, we can state that before the entry into force of FFP, of the two ways indicated by Maqueira et al. (2019) to increase the level of talent on squads, the acquisition of external talent was the main way of seeking to bolster sporting performance, in line with the studies by Rohde and Breuer (2016) and Szymanski (2017), and that these acquisitions were financed through debt. After the implementation of this law, uncontrolled spending is no longer possible, such that clubs are faced with the need to make their management more efficient, so they are opting for the development of internal talent, either to complete their squads, or to sell players and obtain new sources of financing, in addition to the purchase cycles of young players, who, after a few years at the club, are sold for amounts greater than those invested. In recent years, clubs such as Betis, Sevilla, Valencia, Real Sociedad, Athletic Bilbao, Real Madrid, and Barcelona, among others, have been generating income through the sale of players from their youth systems in order to comply with the FFP.

*4.2. Difference between EC and NEC*

To answer the third research question as to whether there are differences before and after the application of FFP to the finances of clubs that participate in European competition and those that do not, the following steps were carried out:

First, we differentiated between two groups of clubs: EC being clubs participating in European competitions, and NEC being clubs not participating in them. We compared the means between the two groups before FFP came into force and after it, and the Mann–Whitney U test was used to verify whether these differences were significant.

Based on the premise of Galariotis et al. (2018) that financial performance influences sporting results, in order to gauge whether FFP could have influenced the competitive balance, we compared whether the differences in the means of the financial statistics of EC vs NEC teams increased or decreased.

Table 2 indicates that all the ratios that were significant were better in the clubs participating in the EC. Thus, although Table 1 shows that the finances of the Spanish league soccer clubs were not in good health, NEC clubs' situations were even more delicate: they were less solvent and less able to generate revenue and cash flows, assets, or quality players (understanding that the higher the intangible assets and personnel expenses, the higher the quality of the squads). The Mann–Whitney U test shows that cash flow, EBITDA, equity, employees, intangible assets, leverage, net turnover, operating income, personnel expenses, PE/OI, PEA, profit, ROA, ROS solvency, and total assets were significant. Though the NEC teams' ROE and RCE figures were better than those of the CE teams, the differences were not significant.

**Table 2.** Testing financial variables based on participation in European Competitions BFFP.

| ITEM | MEAN | | | Mann–Whitney | | |
|---|---|---|---|---|---|---|
| | NEC | EC | Absolute Dif. | Sig | Null | |
| CashFlow | 4,422,014.62 | 26,822,769.66 | 22,400,755. 4 | 0.000 | REJECT | |
| EBIT | −9,696,711.35 | −6,496,666.61 | 3,200,044.75 | 0.495 | | |
| EBITDA | −572,667.29 | 17,751,650.27 | 18,324,317.56 | 0.001 | REJECT | |
| EBT | −10,771,666.79 | −10,421,080.30 | 350,586.50 | 0.878 | | |
| Equity | 2,319,968.97 | 27,909,281.90 | 25,589,312.93 | 0.002 | REJECT | |
| Employees | 103.88 | 217.92 | 114.04 | 0.000 | REJECT | |
| INDEB | 134.97 | 90.03 | −44.93 | 0.124 | | |
| Int Assets | 25,650,929.61 | 68,559,222.82 | 42,908,293.21 | 0.000 | REJECT | |
| LEV | −375.72 | 2,912.89 | 3,288.61 | 0.003 | REJECT | |
| Net tur | 23,482,324.81 | 95,930,909.16 | 72,448,584.35 | 0.000 | REJECT | |
| Oper. Income | 27,951,085.90 | 100,481,721.74 | 72,530,635.84 | 0.000 | REJECT | |
| P. Expen. | 21,858,216.27 | 62,541,271.01 | 40,683,054.74 | 0.000 | REJECT | |
| PE/OI | 84.62 | 68.57 | 16.05 | 0.020 | REJECT | |
| PEA | 241.81 | 314.54 | 72.73 | 0.004 | REJECT | |
| Profit | −4,702,029.45 | 2,574,452.78 | 7,276,482.23 | 0.036 | REJECT | |
| Prof. Marg. | −48.57 | −21.22 | 27.35 | 0.083 | | |
| R. Capital | 9,907,838.86 | 25,488,230.50 | 15,580,391.64 | 0.653 | | |
| RCE | 83.81 | −6.54 | 90.35 | 0.438 | | |
| ROA | −21.79 | −5.14 | 16.65 | 0.042 | REJECT | |
| ROE | 33.88 | −285.39 | 319.27 | 0.341 | | |
| ROS | −156.02 | −27.23 | 128.79 | 0.018 | REJECT | |
| Solvency | 0.58 | 0.75 | 0.16 | 0.002 | REJECT | |
| SRFA | −34.97 | 9.96 | 44.93 | 0.124 | | |
| T Assets | 93,316,055.83 | 298,027,118.87 | 204,711,063.04 | 0.000 | REJECT | |

Table 3 shows the means of the NEC and EC clubs, in addition to the differences between groups and whether or not the null hypothesis was rejected, i.e., whether there were significant differences after FFP. The Mann–Whitney U test shows that the following were significant: cash flow, EBIT, EBITDA, EBT, equity, employees, indebtedness, intangible assets, net turnover, operating income, personnel expenses, PEA, profit, registered Capital, ROE, SRFA, and total assets. The results show that all the significant financial criteria were better for EC teams than NEC ones.

Regarding Research Question 3, as to whether there were differences between the finances of EC and NEC teams before and after the FFP's entry into force, as can be seen in Tables 2 and 3, all the means improved after the implementation of FFP. The column "absolute diff." also shows that the differences between the two groups were *greater* PFFP; despite the overall improvement in Spanish football, clubs participating in the EC are still more solvent, have a greater capacity to generate revenues and cash flows, and have more assets and quality players than BFFP. From the results, it is clear that EC teams have greater financial potential than NEC ones, so there is a relationship between business performance and sporting results, in accordance with the results of Galariotis et al. (2018), and the virtuous circle proposed by Lago et al. (2004) seems to be borne out.

Figure 2 shows those items that are significant both before *and* after FFP's entry into force: cash flow, EBITDA, equity, intangible assets, operating income, net turnover, personnel expenses, profit, and total assets (the employees' and PEA items have been to facilitate viewing of Figure 2). Financially, EC clubs now have a greater capacity to generate revenue, have more resources, and manage them more efficiently. The increase in intangible assets and personnel expenses shows that EC clubs acquire more talent on their squads and pay these players more, without neglecting the rest of the ratios and variables. Thus, there was an increase in the difference between the "financial muscle" of the teams comprising the two groups, taking into account firstly the fact that those participating in EC tend to be a small group of clubs; secondly, that a relationship between business performance and sporting results is observed; and thirdly, that the more talent, the better the sporting

results, the results show that FFP could be exacerbating the competitive imbalance, in line with the findings of Freestone and Manoli (2017); Madden (2015) and Terrien et al. (2017), answering Research Question 3. Therefore, in view of the results, there is evidence to state that there could be a relationship between the sporting results (whether participating or not in European competition) and the financial muscle of Spanish league football clubs.

**Table 3.** Testing financial variables based on participation in European Competitions PFFP.

| ITEM | MEAN | | | Mann–Whitney | |
|---|---|---|---|---|---|
| | NEC | EC | Absolute Dif. | Sig | Null |
| CashFlow | 13,827,578.24 | 58,167,942.30 | 44,340,364.06 | 0.000 | REJECT |
| EBIT | 4,378,329.36 | 21,689,337.18 | 17,311,007.82 | 0.000 | REJECT |
| EBITDA | 15,525,750.87 | 68,224,920.43 | 52,699,169.56 | 0.000 | REJECT |
| EBT | 2,747,020.11 | 16,224,384.01 | 13,477,363.90 | 0.000 | REJECT |
| Equity | 10,033,762.97 | 101,957,187.28 | 91,923,424.31 | 0.000 | REJECT |
| Employees | 175.36 | 435.55 | 260.19 | 0.000 | REJECT |
| INDEB | 133.28 | 76.56 | 56.72 | 0.008 | REJECT |
| Int Assets | 32,696,210.39 | 134,180,988.47 | 101,484,778.08 | 0.000 | REJECT |
| LEV | 209.85 | 238.12 | 28.27 | 0.139 | |
| Net tur | 51,548,372.13 | 238,771,162.82 | 187,222,790.69 | 0.000 | REJECT |
| Oper. Income | 55,413,859.05 | 250,225,914.97 | 194,812,055.92 | 0.000 | REJECT |
| P. Expe. | 37,210,381.03 | 148,182,794.12 | 110,972,413.10 | 0.000 | REJECT |
| PE/OI | 70.39 | 66.77 | 3.62 | 0.379 | |
| PEA | 234.27 | 293.37 | 59.10 | 0.001 | REJECT |
| Profit | 2,738,421.01 | 11,632,359.04 | 8,893,938.03 | 0.002 | REJECT |
| Prof. Marg. | 5.34 | 2.91 | 2.42 | 0.248 | |
| R. Capital | 28,097,762.26 | 83,703,023.71 | 55,605,261.45 | 0.0046 | REJECT |
| RCE | 79.23 | 46.33 | 32.91 | 0.356 | |
| ROA | 8.80 | 3.69 | 5.11 | 0.511 | |
| ROE | 8.86 | 27.07 | 18.21 | 0.001 | REJECT |
| ROS | 2.89 | 5.86 | 2.96 | 0.983 | |
| Solvency | 0.64 | 0.66 | 0.02 | 0.639 | |
| SRFA | −33.28 | 23.44 | 56.72 | 0.008 | REJECT |
| T Assets | 119,916,297.34 | 452,026,486.68 | 332,110,189.34 | 0.000 | REJECT |

Figure 3 shows the means, from 2012 to 2019, of the intangible assets figure, which, among others, includes investment in the purchase of players; personnel expenses, including spending on player salaries; and net turnover. It can be seen that the most successful clubs in Spanish football—FC Barcelona, Real Madrid, and Atlético Madrid, to a lesser extent—are the ones with the most expensive players and the highest salaries. They are also the clubs with the highest revenues, as can be seen in the net turnover item. The change in trend from uncontrolled debt to efficient management has not yet served to reduce the differences, both in terms of financial might and results on the field, between the league's historic juggernauts and the rest of its teams. Based on the data, it seems difficult to overtake these powerhouses of world soccer, since FFP limits the contributions by supporters to be able to undertake investments and expenditures to improve squads. However, it is true, as we have seen throughout the article, that FFP has managed to improve the financial health of the clubs, saving the European soccer industry and, in particular, Spanish soccer. It should be noted that both F.C. Barcelona and Real Madrid, together with Athletic Club de Bilbao, have still not adopted Spain's legal formula of a Sociedad Anónima Deportiva (SAD), or Public Limited Sports Company, so the difference in management between SAD clubs and others could be an interesting line of future research.

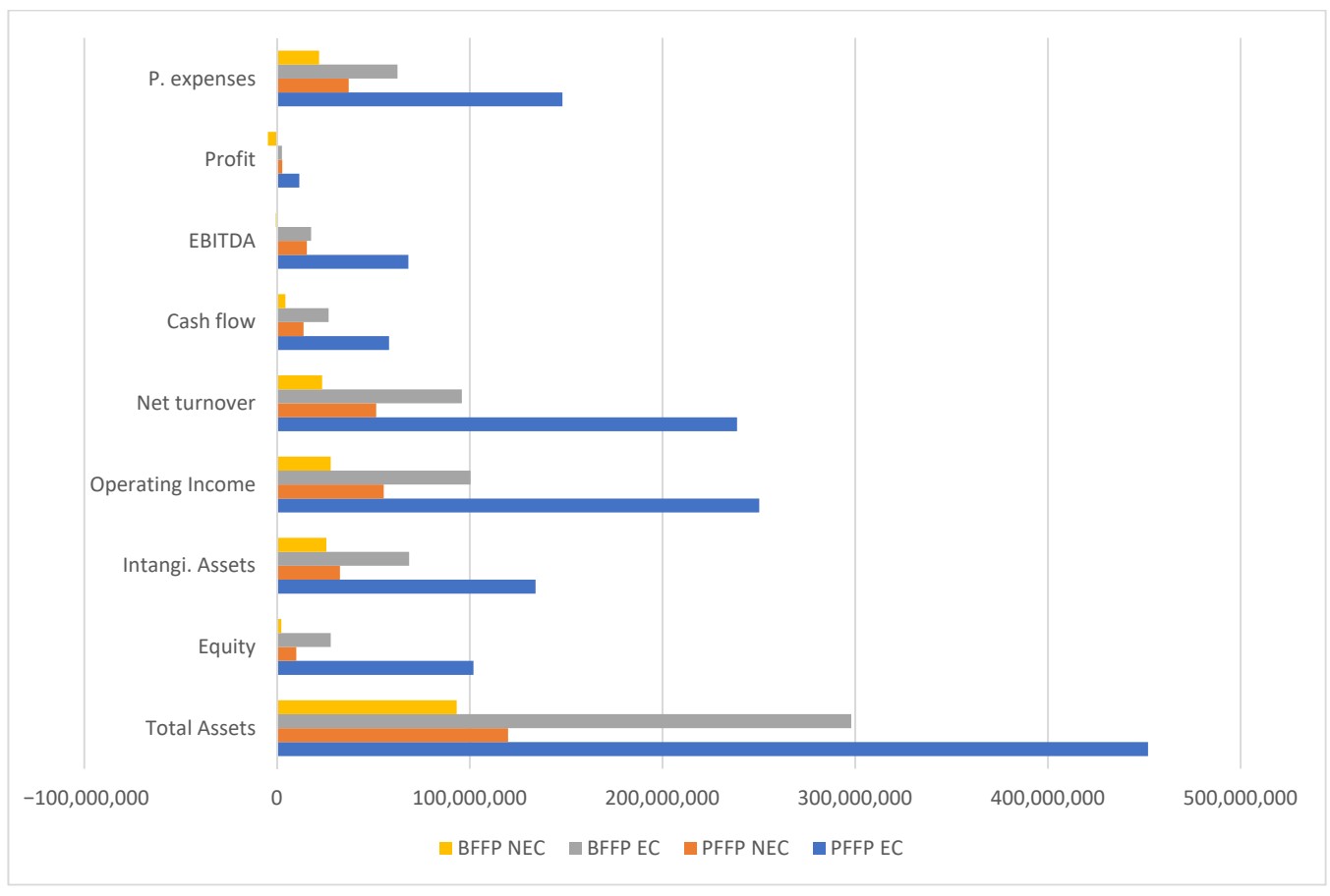

**Figure 2.** The evolution of the significant variables, comparing BFFP and PFFP.

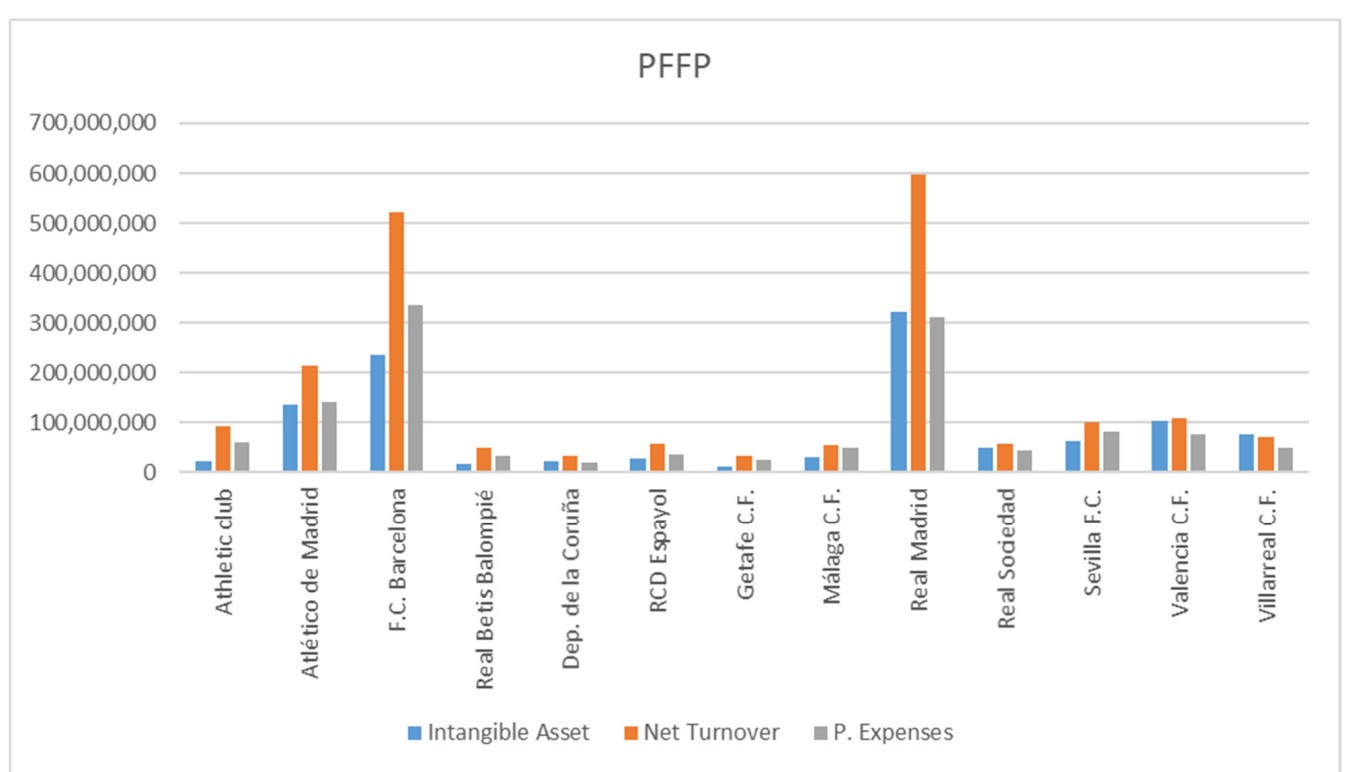

**Figure 3.** Comparative by club PFFP.

## 5. Conclusions

This paper examines the impact of FFP regulations based on the hypothesis that the UEFA, with its new regulatory framework, has largely spawned a shift in the financial efficiency of European football teams, which have gone from engaging in uncontrolled spending to acquire more talent in the pursuit of better sporting results to adopting more efficient models. In order to examine this impact, a number of research questions/issues are proposed and answered in the previous section. The empirical results show that FFP has (1) positively influenced the finances of football clubs, (2) brought about a shift towards a more efficient model, and (3) resulted in differences that can be observed between clubs that participate in European competitions and those that do not (which we defined as sporting objectives).

In view of the results, the paper has shown that, from a financial point of view, the FFP has been a useful tool for improving the medium and long-term solvency of soccer clubs. However, both UEFA and Spanish league managers should not lose sight of the fact that the differences between historically "richer" and "poorer" clubs could actually be increasing, which could exacerbate competitive imbalances, or that we could be facing the falsification of accounting statements to comply with the requirements of the law. In recent years, a very small group of clubs have usually won the Spanish league. Levelling the competitiveness of Spanish soccer and enabling other clubs to compete with Real Madrid and F.C. Barcelona, as Atlético Madrid has managed to, should be one of the priorities of the league to increase interest in La Liga as a product. However, they are the clubs with the highest revenues in the world (Figure 1), and getting them to cede part of this revenue, by relinquishing monies from television rights, in order to favour parity, as has been done in England's Premier League, seems a tall order. The disparity between the top clubs and the rest remains so pronounced that, last summer, the creation of a "European Super League" was proposed, consisting of a type of closed league, resembling the North American models, to feature increased competition between the Continent's dominant clubs. The venture sparked dramatic protest, as it threatened to spell an end of Europe's national leagues. Interesting developments are on the horizon, and the soccer industry could be facing an unprecedented turning point, in a context in which a large number of stakeholders are pursuing different objectives.

In view of the results from the sporting (sports achievements) point of view, FFP demands a "player development" model, as mentioned by several works, such as those by Dimitropoulos and Scafarto (2021), Nicoliello and Zampatti (2016), Pantuso (2017), and Regoliosi (2016), which consists of discovering, recruiting, developing, and selling young players. This model, based on developing sets of young players, increasing their value, and selling them, allows clubs to better conform to the limits imposed by FFP and undertake investments in big-name players, which attract all kinds of sponsors, TV rights, merchandising, etc.

In terms of implications, the paper provides empirical evidence in several respects. Firstly, it has shown that FFP has fulfilled its objective from an economic point of view, as Spanish soccer has shifted from a situation of financial instability to a more sustainable one while still increasing talent acquisition through more efficient management. Taken together, these results could suggest that FFP is driving a shift in the business model of Spanish clubs from one based on uncontrolled spending (Gómez 2017) to a more efficiency-oriented one. The results indicate that clubs have moved away from the uncontrolled talent race towards more sustainable business models, as is the case in the Italian league, according to the work of Dimitropoulos and Scafarto (2021). Secondly, financial efficiency must now be key, as uncontrolled spending and indebtedness are no longer possible, such that financial innovation and efficient management are now crucial.

In terms of implications for UEFA executives, FFP has been the correct financial tool to preserve the financial viability of European football in the medium and long terms, as was its objective. However, UEFA managers should not overlook the fact that in the Spanish league, the difference in "financial muscle" between EC and NEC teams has increased after

the implementation of FFP, as the regulation's limitation on cash contributions by majority shareholders could be aggravating the competitive imbalance between clubs. This could be reinforcing the hierarchies of the top European clubs, as stated in studies by Birkhäuser et al. (2019), Freestone and Manoli (2017), Madden (2015), Sass (2016), and Terrien et al. (2017), at a time when, by means of FFP, UEFA wished to "introduce more discipline and rationality into the finances of club soccer" (UEFA 2012).

As for the implications for club managers, because uncontrolled spending is no longer possible, and due to the aforementioned limitation on monetary contributions by club owners, financial innovation and efficient management become necessary in the search to bolster squads' talent levels (Dimitropoulos and Scafarto 2021). Both external acquisition and internal development produce positive results (Maqueira et al. 2019), but it seems that, after FFP, the acquisition of external talent is not available to all clubs, so the option of internal development must be adopted as a priority in order to comply with legislation and to achieve sporting objectives.

*5.1. Future Research*

Apart from the above, as a future line of research, it would be worthwhile to study how COVID-19 has affected the finances of Spanish soccer. Recently, F.C. Barcelona, in particular, has been in a very delicate financial situation, leading to veritable "financial engineering" to undertake a revamping of the club after the departure of Lionel Messi, on whom much of its economic and sporting plans were based. It has come to the point where the status of the club could be in danger, and there is talk of the possibility of converting it into an SAD as a way to save it. Based on the power of the F.C. Barcelona brand, the good work they have been doing in the development of internal talent since the 1990s, the revenue obtained by this, and their supporters, among others, we believe that the club's managers have several ways to resolve this worrying situation. There should be no need to convert the club into an SAD, but it will take at least two years to right the ship, economically and in terms of its sporting results. In fact, the president of F.C. Barcelona, Joan Laporta, proposed to the club's members now-famous "economic levers" at its extraordinary assembly in 16 June 2022, i.e., to assign or dispose of part of the club's assets to obtain revenue that will allow him to comply with FFP (Soldevilla 2022). These "levers" were approved by the delegate members, with which it will be possible to raise between 700 and 800 million EUR. The sales will consist of 49.9% of BLM (Barça Licensing and Merchandising) and up to 25% of La Liga television rights (De la Rosa 2022). Therefore, it is necessary to study the causes that brought the club to this situation, whether due to mismanagement of its resources, the consequences of COVID-19, or the loss of the cornerstone of its plans in recent years.

It is not only F.C. Barcelona that is experiencing financial problems. In recent years, there has been a loss of talent in the Spanish league with respect to those of other nations, such as England, Germany, Italy, and France, which are surpassing the Spanish clubs in the acquisition of talent. The latest example was the renewal of Kylian Mbappé by Paris Saint Germain. In statements made by Real Madrid's own president on a television programme on the night of 15–16 June 2022, the club was not able to compete with the economic terms offered to the player. Another example is Valencia C.F., historically one of Spain's major soccer clubs, which in recent years has been forced to sell its top stars in order to recover from an adverse economic situation going back years. As a future line of research, it is necessary to understand the causes underlying this loss of talent in the Spanish league, which we believe could be due to the fact that the application of the FFP regulation in Spain has differed significantly from the way it has been applied in the rest of the major football leagues, which may have resulted in unfair competition and an ensuing loss of competitiveness. In fact, Spanish Soccer Federation President Javier Tebas recently filed a complaint with UEFA for alleged non-compliance with FFP by Manchester City and Paris Saint Germain, in the form of capital injections by third parties (Sánchez 2022).

In addition to the above, one of the causes of this loss of talent could be Spain's onerous taxation terms. Thus, it is necessary to address tax reform, especially following the

elimination of the application of the Expatriate Regime, also known as the "Beckham Law", in 2010, formerly covering all professional athletes residing in Spain, according to which tax rates on income earned in Spain ranged from 19% to 24.75%. With the rescinding of the law, the top rate shot up to 50%. In Italy, for example, following the implementation of a measure announced in 2016 and which came into force with the 2017 Budget Law, the taxation of certain income from outside Italian territory is capped at 100,000 EUR. In addition, its tax regime for expatriates allows foreign footballers who establish residence in the country to benefit from a tax exemption on 70% of the income they receive from the club, with this ranging up to 90% if they establish their residence in southern Italy (Zambrano-Domínguez 2022). All this points to the disadvantages of Spanish clubs with respect to other countries, such that we see the taxation applied by different countries, and its impact on the talent of their soccer clubs, to be another interesting line of future research. Lastly, it would be valuable to analyse how the different financial/accounting ratios impact the probability of achieving sporting objectives and how the entry of FFP has modified this influence.

*5.2. Limitations*

The limitations of the study include that its sample consists of clubs from a single league: La Liga. As was explained above, the Spanish league is special because it has no stock market presence but still manages to stand as that with the highest revenue. Second, this study does not control for the possibility of clubs resorting to potentially deceptive or illicit practices in the preparation of their financial statements to comply with FFP. Thirdly, other limitations are related to the statistical analyses selected, such as the possibility of Type-I errors due to distinct variances. In order to avoid Type-I errors, the data set was tested for various turning points. While the authors do not expect this limitation to affect this work, it cannot be disregarded entirely.

**Author Contributions:** Conceptualization, A.C.-L., M.R. and J.V.-O.; methodology, A.C.-L., M.R. and J.V.-O.; software, J.V.-O.; validation, A.C.-L., M.R. and J.V.-O.; formal analysis, A.C.-L., M.R. and J.V.-O.; investigation, A.C.-L.; resources, A.C.-L.; data curation, A.C.-L. and J.V.-O.; writing—original draft preparation, A.C.-L. and M.R.; writing—review and editing, M.R.; visualization, A.C.-L., M.R. and J.V.-O.; supervision, M.R.; project administration, A.C.-L. and M.R.; funding acquisition, A.C.-L. All authors have read and agreed to the published version of the manuscript.

**Funding:** This research received no external funding.

**Informed Consent Statement:** Not applicable.

**Data Availability Statement:** Data are available in SABI.

**Acknowledgments:** The authors are grateful to the participants in the "XI AECA International Conference on Valuation, Financing and Risk Management" (2022). We also acknowledge the financial support of the University of Jaen (SEJ-289 research group: Information and Management Systems in Andalusian Enterprises).

**Conflicts of Interest:** The authors declare no conflict of interest.

**Appendix A**

**Table A1.** Variables calculation.

| Ratios | Abbreviation | Calculation | Source |
|---|---|---|---|
| Indebtedness | Indeb | (total liabilities − equity)/total liabilities × 100 | SABI |
| Leverage | Lev | (non-current liabilities + financial liabilities)/equity × 100 | SABI |
| personal expenses/operating income | PE/OI | personal expenses/operating income | SABI |

| Ratios | Abbreviation | Calculation | Source |
|---|---|---|---|
| Average personal expenses | PEA | personal expenses/number of employees | SABI |
| Profit margin | Prof. Marg | EBT/operating income | SABI |
| Registered capital | R. Capital | | SABI |
| return on capital employed | RCE | (EBT + Finance expenses and assimilated expenses)/(equity + non-current liabilities) $\times$ 100 | SABI |
| Return On Assets | ROA | (EBT/total assets) $\times$ 100 | SABI |
| Return on Equity | ROE | (EBT/equity) $\times$ 100 | SABI |
| Return on sales | ROS | (EBIT/net turnover) $\times$ 100 | AUTHORS |
| Solvency ratio | Solvency | (current assets/current liabilities) | SABI |
| ratio of financial autonomy | SRFA | (equity/total assets) | SABI |

## Note

[1]   The participation of clubs in both the top division and in European competitions was verified on the official website of the Professional Football League (LFP).

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
