# Peer review of "The Financial/Accounting Impact of FFP on Participating in European Competitions: An Analysis of the Spanish League"

_ijfs, doi:10.3390/ijfs10030081_

Round 1

Reviewer 1 Report

The paper is generally well written and there is appropriate engagement with academic literature.  However, there are two substantial areas that need to be revised.  I also raise some comments in relation to the methodology too.  

First, at various points (abstract, introduction) it is stated that the study will show the relationship between financial performance and sporting results.  Apart from some mention of this in the literature review this is absent from the empirical results that is presented.  If indeed the study is only investigating the financial impact of FFP, as implied by the title, then all reference to financial performance and sporting results should be removed.

Second, the conclusion section is way too long.  Much of this material needs to be incorporated into the empirical results and associated discussion section.

As it stands this paper closely follows the paper by Dimitropoulos and Scafarto (2021).  You should stipulate why the methodology from this paper was followed and state what changes / improvements (if any) were made.  I would also like to see some sensitivity analysis in the way the before and after FFP was determined.  For example, what happens if the post FFP period is changed?  Your study has chosen 2011-2012 for the post period but are the results affected had you chosen, for example, 2012-2013 or 2013-14 or 2014-15 instead?

The definitions of some of the financial measures listed inthe appendix needs to be clearer.  For example, you say profit (loss) but is this net, gross or operating profit (loss)?  Fully define ROA and ROE - don't just say these are economic and financial profitability measures respectively say what the actually are (e.g. ROA return on assets). 

Reviewer 2 Report

The paper's idea is exciting and potentially sheds light on financial management in the sports industry and can be easily applied to other sports than soccer as well.

I have the following suggestions and hope they will help to improve the overall quality of the paper:

1. The evolution of sports regulation in Europe and Spain can be moved to the introduction section to show why this study is important;

2. Methodology: the authors only conducted univariate analyses of the data they collected, which is not significant enough for publication. I would recommend the authors consider multi-variant regression models to examine the pre- and post-FFP regulation on the financial performance and competitiveness of the teams. And if financial performance can predict the sports performance of a team after this regulation.

3. Conclusion needs to be more concise.

Reviewer 3 Report

1- Authors have done a great job with this evolving topic. 

2- I suggest that authors explain the term FFP before starting to use it in the first sentence in the Abstract. 

3- The methodology is not clear.

4- Conclusion is way to long and can be spread out by adding a heading or Future Recommendations. 

5-Increase the references/literature review

Round 2

Reviewer 2 Report

Thank you for your hard work in reorganizing the paper, but it needs to be copy-edited by a professional writer. There are lots of errors and mistakes in writing.

Author Response

Dear Reviewer,

thank you for your valuable comment. The text has been copy-edited by a professional writer. Thank you for your suggestions and contributions, which have undoubtedly improved the quality of the paper.

Sincerely yours, Alberto Calahorro-López, on behalf of my co-authors.

Round 3

Reviewer 2 Report

Nice job on copy-editing the paper.